# Fast Training Dataset Attribution via In-Context Learning

**Milad Fotouhi** [1]  **Taha Bahadori** [1]  **Seyi Feyisetan** [1]  **Payman Arabshahi** [2]  **David Heckerman** [1]

## Abstract

We investigate the use of in-context learning and prompt engineering to estimate the contributions of training data in the outputs of instruction-tuned large language models (LLMs). We propose two novel approaches: (1) a similarity-based approach that measures the difference between LLM outputs with and without provided context, and (2) a mixture distribution model approach that frames the problem of identifying contribution scores as a matrix factorization task. Our empirical comparison demonstrates that the mixture model approach is more robust to retrieval noise in in-context learning, providing a more reliable estimation of data contributions.

## 1. Introduction

Training Data Attribution (TDA) refers to the task of quantifying contributions of different data sources on the outputs of a model (Park et al., 2023; Nguyen et al., 2023). This task is essential for debugging the processes of curating corpora for training and for improving the training of neural networks. Understanding the contribution of data sources allows us to assess the monetary value of proprietary training data, which is crucial for fair compensation and data management (Ghorbani & Zou, 2019; Nohyun et al., 2022).

Existing methods for TDA, primarily fall into two categories: retraining-based methods and influence function-based methods, as detailed in recent surveys (Hammoudeh & Lowd, 2024; Worledge et al., 2024). Retraining approaches such as those by (Feldman & Zhang, 2020; Ghorbani & Zou, 2019) involve retraining the model without the target data source. However, this method is computationally expensive. Influence function approaches (Koh & Liang, 2017; Pruthi et al., 2020; Chen et al., 2021; Park et al., 2023), relax the need for full retraining by requiring only a few gradient calculations with respect to the data. Despite their

efficiency, these methods rely on a linear approximation of the neural network around the target data point, which can be inaccurate. Critically, the influence function approaches compute the attribution score for a dataset as a linear function (usually an average or sum) of the attribution scores for each data point in the dataset (Hammoudeh & Lowd, 2024; Park et al., 2023). This approach fails to provide a holistic view of the contributions of an entire dataset to the model's output. additionally, both methods require access to the internals of LLMs, which is not feasible for some popular models.

We explore the use of in-context learning and prompt engineering to estimate the contributions of each dataset as a whole in the outputs of instruction-tuned LLMs. We propose two approaches: (1) A similarity-based approach, which posits that providing a dataset as context to an LLM trained on that dataset changes its output less compared to when the LLM was not trained on the dataset. (2) A mixture distribution model approach, where we model the behavior of LLMs using a new mixture distribution. This approach transforms the problem of identifying contribution scores into a matrix factorization problem, which we solve using the alternating projected least squares method. Both approaches utilize Retrieval Augmented Generation (RAG) (Lewis et al., 2020) to accommodate large data sources.

In the experiments, we evaluate four instruction-tuned LLMs: Mistral 7B (Jiang et al., 2023), Bloomz (Le Scao et al., 2023), Microsoft/Phi-3-mini (Abdin et al., 2024) and GPT 4.0 (Achiam et al., 2023) on a set of binary Q&A datasets, `boolq` (Clark et al., 2019). We demonstrate that the mixture model approach is more robust to the retrieval noise inherent in RAG systems, providing more accurate estimations of data contributions. Additionally, we conduct an evaluation using Trak (Park et al., 2023) to compare our results with those obtained by the Trak method, validating the effectiveness and efficiency of our proposed approaches..

## 2. Methodology

An LLM stores knowledge from different sources. Our goal is to examine different prompts and see if we can uncover the sources of this knowledge.

In the binary outcome setting, we have tuples in the format

*Equal contribution [1]Amazon [2]University of Washington. Correspondence to: Milad Fotouhi <mfotouhi@amazon.com>.

of question, context, and outcome: $(q, c, y)$. When we don't use any context, we denote $c = \emptyset$. Our model also outputs $p(y|q, c)$. Our goal is to quantify the contributions of the training datasets $\mathcal{D}_1, \ldots, \mathcal{D}_n$ in $p(y|q, c)$. We assume that we have a query set $Q = \{q_1, \ldots, q_m\}$.

We have access to probabilistic distances $d(p_1, p_2)$ where $p_1$ and $p_2$ are the probabilities obtained by two different prompting mechanisms. The distance can be common distance metrics such as KL divergence, total variation, or Wasserstein distances.

We assume that we have $k, k = 1, \ldots, K$ relevant datasets about a topic and we want to quantify their contributions in the generation of the output by our LLM.

## 2.1. Approach #1: The Shapley Context Method (SCM)

The key idea of this approach is that if an LLM uses the information from the $k$th dataset, providing the $k$th dataset as context will not change the output much. Conversely, if adding a dataset as context changes the output significantly, it was likely not used for generation of the output.

$$s_k = \text{sim}(y, y|c_k), \tag{1}$$

where $c_k$ is the context provided from the $k$th dataset.

Usually, desired information can be found in multiple datasets (Ghorbani & Zou, 2019). To consider the impact of datasets in presence of other datasets, we define the following residual scores to be used in the Shapley formula (Shapley, 1953):

$$s_S = \text{sim}(y, y|c_S). \tag{2}$$

The Shapley values are computed as follows:

$$\phi_k = \sum_{S \subseteq \{D_1, \ldots, D_K\} \setminus \{D_k\}} \frac{|S|!(K - |S| - 1)!}{K!} \left(s_{S \cup \{k\}} - s_S\right). \tag{3}$$

This formula finds the residual increase in the similarity by including $D_k$, when we already have included another set $S \subseteq \{D_1, \ldots, D_K\} \setminus \{D_k\}$. Algorithm 1 in Appendix B describes the details of our Shapley Context Method (SCM).

## 2.2. Approach #2: Context Mixture Factorization (CMF)

We propose a model for summarizing the behavior of LLMs. Our model explicitly contains attribution scores and captures the entirety of the datasets used for its training. We use a mixture distribution approach, which defines:

$$p(y|q) = \pi_0 p_0(y|q) + \sum_{k=1}^{K} \pi_k p_k(y|q), \tag{4}$$

where $p_0$ denotes a general-purpose language model and $p_k$ denote the language models specialized on each of the relevant datasets $k = 1, \ldots, K$.

*Remark:* Given the modularity of LLM structures, this assumption is not fully realistic. However, this assumption provides a holistic view of the contributions of each dataset, captured by distributions $p_k, k = 1, \ldots, K$. Thus, Model (4) serves as a useful tool to statistically summarize the behavior of the LLM.

We model the impact of providing context from a dataset $k \in \{1, \ldots, K\}$ as an intervention in the probability distribution:

$$p(y|q, c_k) = \pi_0 p_0(y|q) + (1 - \pi_0) p_k(y|q). \tag{5}$$

The key assumption is that both Eq. (4) and (5) do not have context terms in the right-hand side quantities. See a weaker version of Assumption (5) in the appendix.

**Goal:** Our goal is identifying $\pi_k, k = 1, \ldots, K$. Which probability distance metrics help identify these contributions? We want to do this without explicitly estimating $p_k, k = 1, \ldots, K$.

**Formulating as a Matrix Factorization Problem**

For each of the $m$ queries, we perform $K + 1$ prompts (or $2^K$ prompts) and write the results in a linear equation as follow:

$$\boldsymbol{p}^{(j)} = \Pi \boldsymbol{p}_k^{(j)}, \quad \text{for} \quad j = 1, \ldots, m.$$

We observe the left-hand side, but none of the quantities in the right-hand side. The matrix $\Pi$ has a specific structure. By defining the matrix $P = [\boldsymbol{p}^{(1)}, \ldots, \boldsymbol{p}^{(m)}]$, we can write our problem in the following matrix form:

$$P = \Pi P_K, \tag{6}$$

where $P \in [0, 1]^{(K+1) \times m}$, $\Pi \in [0, 1]^{(K+1) \times (K+1)}$, and $P_K \in [0, 1]^{(K+1) \times m}$.

This is a matrix factorization problem with a special structure. We assume that you can obtain $p_k(y|q)$ via some clever prompts. We can make assumptions about $p_k(y|q)$ that allows recovery of $\pi_k$ parameters.

*Remark* Instead of $K + 1$ prompts, we can have $2^K$ prompts. However, for the prompts that use multiple datasets, we need to assume the form of the resulting distribution, similar to Eq. (5). An alternative is to put priors on $\boldsymbol{\pi}$ and $P_K$ to improve identifiability. We pursue the second approach in the next section.

**Alternating Projected Least Squares**

We can have multiple estimates for $\boldsymbol{\pi}$ from Eq. (6). We can resolve this issue be encouraging solutions that have

lower variance. We achieve this by using two regularizers: an entropy regularizer for $\boldsymbol{\pi}$ to assume that the sources contribute equally and a regularizer that encourages $P_K$ to be less informative.

$$\widehat{\boldsymbol{\pi}} = \arg\min_{\boldsymbol{\pi}} \min_{P_K} \|P - \Pi P_K\|_F^2 \qquad (7)$$
$$- \lambda_\pi H(\boldsymbol{\pi}) + \lambda_{P_K} \|P_K - 1/2\|_F^2,$$
$$\text{s.t.} \qquad \boldsymbol{\pi} \succeq \mathbf{0}, \ \mathbf{1}^\top \boldsymbol{\pi} = 1$$
$$0 \preceq P_K \preceq 1,$$

where $\|\cdot\|_F$ and $H(\cdot)$ denote the Frobenius norm and Shannon's entropy. We use entropy regularization on $\boldsymbol{\pi}$ encouraging the null hypothesis of "equal contributions of all sources". The Frobenius norm regularization implies that unless there is strong evidence, the outputs of the latent probabilities $P_K$ should be $1/2$. Note the regularizers are vital for obtaining a non-trivial solution, and in absence of the them, there are many solutions for the problem.

The problem in Eq. (7) is biconvex; i.e., fixing either of $\boldsymbol{\pi}$ or $P_K$, the problem is convex. Thus, we solve it by the alternating least squares method. We describe the procedure in Algorithm 2 in Appendix B. In Appendix A we study the impact of regularization coefficients on the uncertainty of the estimation using synthetic datasets.

## 3. Preliminary Experiments

We utilized the BoolQ Q&A dataset for evaluating the models, focusing on binary Yes/No answers. Through prompt engineering, we developed a prompt instructing the models to answer directly with "Yes," "No," or "I don't know." This approach yielded improved responses for GPT-4, Bloomz, and Mistral 7B but required additional measures for Phi-3-mini, for which we devised a Zero-shot classification layer to measure similarity more accurately.

Given the impracticality of fitting entire datasets into the context windows of LLMs, we employed Retrieval-Augmented Generation (RAG) to enhance the context by retrieving relevant documents. This involved splitting documents into meaningful chunks, computing embeddings, and storing them in a vector database. During query processing, the most relevant documents were retrieved and provided as context to the LLMs, enabling them to generate more informed responses. We provide further implementation details in Appendix C.

Simplified setup to demonstrate our methodology:

**Step 1: Task Selection** We use the BoolQ Q&A dataset, which consists of tuples in the form (question, relevant context, binary answer) for each question.

**Step 2: LLM Selection** We examined four instruction-tuned LLMs: GPT-4, Bloomz, Mistral 7B, and Phi-3-mini.

| Metric | Bloomz | GPT-4 | Mistral 7B | Phi-3 |
|---|---|---|---|---|
| $\phi_{\text{BoolQ}}$ | 0.48 | 0.59 | 0.57 | 0.50 |
| $\phi_{\text{Chemistry}}$ | 0.10 | 0.08 | 0.09 | 0.10 |
| $\phi_{\text{Natural Science}}$ | 0.12 | 0.09 | 0.10 | 0.11 |
| $\phi_{\text{History}}$ | 0.11 | 0.10 | 0.10 | 0.11 |
| $\phi_{\text{Biology}}$ | 0.10 | 0.07 | 0.08 | 0.10 |
| $\phi_{\text{Law}}$ | 0.09 | 0.07 | 0.08 | 0.08 |

*Table 1.* Shapley Values ($\phi_k$) using SCM Algorithm

We report the accuracy of these LLMs on the Q&A task in Table 5. Given that the dataset is binary, we prompted the LLMs to answer "Yes" or "No" to each question, or to say "I don't know" if they could not provide a definite response (see Section C.1).

**Step 3: Dataset Collection** We collected five datasets on different topics. The corpora were sourced from a subset of the Wikipedia Field of Science dataset available on Hugging Face, specifically the fields of Chemistry, Natural Science, History and Archaeology, Biology, and Law. Each of these datasets contains more than a million samples across five categories.

**Step 4: Evaluation** Clearly, the context from BoolQ is more related to the questions. Thus, the successful methods should estimate higher weight for BoolQ, *as the proxy for the relevant data* used during training. For Approach #1, we report $\phi_k$ for 6 sources. Similarly, we report 7-dimensional $\boldsymbol{\pi}$ vector for Approach #2. The first dimension $\pi_0$ represents the contribution of all other data sources.

**Results and Analysis** Tables 1 and 3 show the attribution results obtained by SCM and CMF algorithms, respectively. Both algorithms successfully identify the BoolQ dataset as the most influential dataset. This is because BoolQ context is more directly related to the questions. Chemistry, Natural Science, History and Archaeology, Biology, and Law have lower $\phi_k$ values, showing that while they contribute to the context, their impact is less significant compared to BoolQ. Note that in CMF, we need to calculate $\frac{\pi_{\text{BoolQ}}}{1 - \pi_{\text{Base}}}$ to directly compare it with $\phi_{\text{BoolQ}}$ estimated by SCM.

Comparing the results of SCM and CMF, we see that CMF places higher weight on BoolQ. With all LLMs, the quantity $\frac{\pi_{\text{BoolQ}}}{1 - \pi_{\text{Base}}}$ is larger than $\phi_{\text{BoolQ}}$ obtained by SCM. We attribute this to the noise in RAG when we pull context from multiple sources. Moreover, CMF provides the contributions of the base LLM $\pi_{\text{Base}}$. We can see that GPT-4 has the highest contribution from the base language model. This observation is inline with the observation that GPT-4 has the highest accuracy without any context, as reported in Table 5 in the appendix.

We also conducted an evaluation using Trak (Park et al.,

2023) as a popular baseline. Trak provides a different perspective on dataset attribution by scoring the impact of training data on model predictions. The Trak scores for Bloomz and Phi-3-mini across the same datasets are shown in Table 2 . Our results align well with Trak, and improve upon it, reinforcing the robustness of our methods.

| Model | BoolQ | Chemist | Natural Sci | History | Biology | Law |
|---|---|---|---|---|---|---|
| **Phi-3** | 0.58 | -0.08 | 0.20 | 0.18 | -0.05 | 0.07 |
| **Bloomz** | 0.61 | -0.10 | 0.22 | 0.19 | -0.06 | 0.09 |

*Table 2.* TRAK Scores for Different Sources using Phi-3 and Bloomz models. Positive scores indicate datasets that contribute positively to the model's output, while negative scores indicate a lesser or inverse influence.

By comparing our results with Trak, we observe that both methods identify BoolQ as the most significant contributor. However, our CMF approach provides a more detailed and accurate attribution of dataset contributions, particularly in quantifying the base model's influence and managing the noise inherent in RAG systems.

**Runtime Comparison** The CMF algorithm is faster than the SCM algorithm as it requires fewer queries with shorter context sizes. Utilizing an AWS EC2 G6 instance (g6.16xlarge), the total runtime for CMF, involving 7 runs, ranges from 77 to 94 minutes for all LLMs. In contrast, the SCM method, which requires $2^5$ runs, results in a total runtime of 352 to 384 minutes. Both algorithms' runtimes are dominated by the RAG search time. This substantial reduction in runtime demonstrates the efficiency of the CMF method, making it more suitable for scenarios demanding both accuracy and computational efficiency.

For TRAK, the runtime is significantly higher due to its high memory requirements. TRAK requires about 20 GB of GPU memory for a model with 1 million parameters. Scaling this to larger models, TRAK's memory requirements become impractical for large LLMs with modest computing resources. Running TRAK on our LLMs would necessitate approximately 600 GB of GPU memory and significantly more computational time, making CMF and SCM more feasible for our use case.

**Deep Dive into RAG Noise Effect** We compute the mean similarities and residuals for each model as shown in Table 4. The high residual for Bloomz (-0.32) indicates that BoolQ contributes significantly to the output. When the BoolQ context is added, the model's performance changes markedly, demonstrating the influence of this dataset.

The low residual for GPT-4 (-0.03) suggests that GPT-4 has been pre-exposed to similar data. The minimal change in performance with the addition of BoolQ context indicates that GPT-4 already possesses substantial knowledge from

| Metric | Bloomz | GPT-4 | Mistral 7B | Phi-3 |
|---|---|---|---|---|
| $\pi_{\text{Base}}$ | 0.05 | 0.08 | 0.06 | 0.05 |
| $\pi_{\text{BoolQ}}$ | 0.60 | 0.62 | 0.61 | 0.59 |
| $\pi_{\text{Chemistry}}$ | 0.09 | 0.07 | 0.08 | 0.09 |
| $\pi_{\text{Natural Science}}$ | 0.07 | 0.08 | 0.07 | 0.06 |
| $\pi_{\text{History}}$ | 0.08 | 0.10 | 0.09 | 0.07 |
| $\pi_{\text{Biology}}$ | 0.06 | 0.06 | 0.05 | 0.05 |
| $\pi_{\text{Law}}$ | 0.05 | 0.10 | 0.08 | 0.06 |
| $\frac{\pi_{\text{BoolQ}}}{1-\pi_{\text{Base}}}$ | 0.63 | 0.67 | 0.65 | 0.62 |

*Table 3.* $\pi$ and $\frac{\pi_{\text{BoolQ}}}{1-\pi_{\text{Base}}}$ values using CMF algorithm

similar datasets.

The positive residual for Mistral 7B (0.06) implies that Mistral 7B relies on the added context. This reliance on context suggests that the model benefits greatly from the additional information provided by BoolQ, indicating that it has not been exposed to similar data during training. However, this also introduces RAG noise, as the model's output can be significantly influenced by the context.

The small residual for Phi-3 (0.03) shows partial exposure to similar data during training. The model benefits from the added BoolQ context but to a lesser extent, suggesting that while it improves with additional context, it already has some degree of relevant knowledge.

| Metric | Bloomz | GPT-4 | Mistral 7B | Phi-3 |
|---|---|---|---|---|
| $s_S$ | 0.86 (0.03) | 0.76 (0.02) | 0.73 (0.03) | 0.60 (0.03) |
| $s_{S\cup\{D_k\}}$ | 0.54 (0.02) | 0.72 (0.02) | 0.69 (0.03) | 0.63 (0.02) |
| $r_{k,S}$ | -0.32 (0.02) | -0.03 (0.02) | 0.06 (0.02) | 0.03 (0.02) |

*Table 4.* Mean similarities $s_S$ and residuals $r_{k,S}$ for different LLMs (with standard deviations)

## 4. Conclusion and Discussion

Our results show that both of our proposed algorithms successfully attribute the output of LLMs to the BoolQ dataset (as the proxy for related knowledge). Comparing LLMs GPT-4 showed minimal change in the similarities when BoolQ context was added, suggesting prior exposure to similar data, while Bloomz exhibited a high residual, indicating substantial influence from BoolQ. The CMF algorithm provides further insights by quantifying the contributions of the base LLM. Comparing our two methods, we conclude that CMF is computationally less expensive and more robust to the RAG noise.

In this paper, we considered several datasets as proxies for the datasets used for training of LLMs. For future work, we will try to avoid this approximate method and train our LLMs on specific datasets and test our algorithms with ground truth contributing datasets obtained by retraining.

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

# A. Synthetic Experiments with Matrix Factorization

To understand the impact of regularization terms in Eq. (7), we perform a simple synthetic experiment. We create a matrix of random $p(y|q)$ (uniformly between 0 and 1) for $m = 5$ queries and decompose it to identify the impact of 3 sources. We vary the regularization terms $\lambda_\pi = \lambda_{P_K} \in [0, 10^{-4}, 10^{-3}, 10^{-2}, 10^{-1}, 1]$. We repeat the experiments with randomly initialized $bm\pi$ and $P_K$ for 100 times and report the variations in the solutions for $\pi$. The variation is measured as the sum of the eigenvalues of the covariance matrix of the solutions (i.e., nuclear norm). Figure 1 recommends choosing the regularization coefficients larger than $0.1$ to have stable solutions.

# B. Algorithms

# C. Implementation Details

## C.1. Prompt Engineering

For simplicity of evaluation and without loss of generality, we used BoolQ (Clark et al., 2019) Q&A dataset, where the answers are binary Yes/No. To instruct the LLMs to provide direct boolean responses, we used prompt engineering. Initially, we tested various prompts without explicitly instructing the model to answer with "Yes" or "No." Diverse examples used in this process are provided in Appendix C.3. Through iterative testing, we found that responses improved when the model was explicitly instructed to provide a boolean answer. This led to our final prompt:

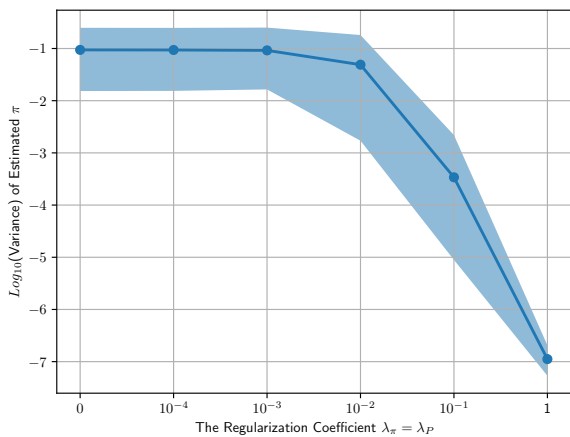

*Figure 1.* The Logarithm of the variations in the solutions for $\boldsymbol{\pi}$. The variation is measured as the sum of the eigenvalues of the covariance matrix of the solutions (i.e., nuclear norm). The shaded area shows the [5, 95] percentile.

**Prompt:** "Given the context below, answer the question that follows with only 'Yes', 'No', or 'I don't know' if the context is insufficient.
{question}? The answer to this question is"

While this final prompt worked well for GPT-4, Bloomz, and Mistral 7B, generating straightforward "Yes," "No," or "I don't know" responses, it was harder to instruct Phi-3-mini. Even with the final prompt, Phi-3-mini often generated more text than just a simple boolean response.

Therefore, calculating similarities was straightforward for GPT-4, Bloomz, and Mistral 7B, but we had to devise another solution for Phi-3-mini. The embedding similarity API on GPT-4 was not precise enough as it did not focus primarily on the context of the generated response. To calculate the similarity for Phi-3-mini, we created a Zero-shot classification layer (which takes 1000 characters) between the prediction and the result to measure similarity more accurately.

### C.2. Using RAG

Given the limitations of LLM context windows, fitting entire datasets directly into the context is impractical. To address this, we utilized Retrieval-Augmented Generation (RAG) (Lewis et al., 2020) to enhance the context by retrieving relevant documents from databases before generating responses. The process involves splitting the documents into semantically relevant chunks using the *RecursiveCharacterTextSplitter* from the Hugging Face Transformers library, computing embeddings for all chunks with a model like *thenlper/gte-small*, and storing these embeddings in a vector database using FAISS (Facebook AI Similarity Search). When a question is posed, it is embedded and a similarity

---

**Algorithm 1** Shapley Context Method (SCM)

**Require:** An instruction-tuned LLM $M$ that generates outputs $y$ for each query $q$ and context $c$.
**Require:** A set of queries $Q = \{q_1, \ldots, q_m\}$.
**Require:** A set of datasets that we need to compute their contributions $D_1, \ldots, D_K$.
1: **for** $q \in Q$ **do**
2:     Compute output without context: $y = M(q)$.
3:     **for** $S \subset \{D_1, \ldots, D_K\} \setminus \{D_k\}$ **do**
4:         Use RAG to create contexts $c_S$ and $c_{S \cup \{D_k\}}$ from the datasets in $S$ and $D_k$.
5:         Compute the output with the context $y|c_S = M(q|c_S)$.
6:         Compute the output with the context including $D_k$: $y|c_{S \cup \{D_k\}} = M(q|c_{S \cup \{D_k\}})$.
7:         Compute the similarities $s_S$ and $s_{S \cup \{k\}}$ using Eq. (2).
8:         Compute the residual gain $r_{k,S} = s_{S \cup \{k\}} - s_S$.
9:     **end for**
10:    Use Eq. (3) and $r_{k,S}$ to compute $\phi_k(q)$.
11: **end for**
12: **Return** average $\phi_k$ over $m$ queries.

---

search is performed against the vector database to find the closest matching documents. These retrieved documents are then provided as context to the LLMs along with the original question, allowing the LLMs to generate responses augmented with additional context.

We used a chunk size of 512 and a top-k value of 3, ensuring the context was trimmed to 2000 characters for conciseness.

### C.3. Prompts

**General Question Prompt:** "Read the context provided and answer the following question: {question}"

**Contextual Understanding Prompt:** "Based on the information in the context, what can you conclude about the following question? {question}"

**Summarization Prompt:** "After considering the context below, summarize your answer to this question: {question}"

**Opinion-Based Prompt:** "Given the details in the context, what is your opinion on the following question: {question}"

**Detail Extraction Prompt:** "Extract relevant information from the context to answer this question: {question}"

**Fact-Checking Prompt:** "Using the context provided, verify the accuracy of the following statement: {question}"

Table 5 shows the average accuracy calculated by comparing the predictions with the ground truth from BoolQ.

---

**Algorithm 2** Context Mixture Factorization (CMF)

---

**Require:** An instruction-tuned LLM $M$ that generates outputs $y$ for each query $q$ and context $c$.
**Require:** A set of queries $Q = \{q_1, \ldots, q_m\}$.
**Require:** A set of datasets that we need to compute their contributions $D_1, \ldots, D_K$.
 1: **for** $q \in Q$ **do**
 2:     Compute output without context: $p[y|q, c_0] = M(q)$.
 3:     **for** $k = 1, \ldots, K$ **do**
 4:         Use RAG to create context $c_k$ from the dataset $D_k$.
 5:         Compute the output with the context $p[y|q, c_k] = M(q|c_k)$.
 6:     **end for**
 7: **end for**
 8: Build matrix $P$, where $P_{k,j} = p[y|q_j, c_k]$.
 9: Solve Eq. (7) via alternating least squares and to compute $\widehat{\pi}$.
10: **Return** the contribution vector $\widehat{\pi}$.

---

| Context | Bloomz | GPT-4 | Mistral 7B | Phi-3 |
|---|---|---|---|---|
| No Context | 0.43 | 0.73 | 0.68 | 0.70 |
| BoolQ as RAG | 0.74 | 0.87 | 0.85 | 0.82 |
| Five Datasets Only | 0.35 | 0.64 | 0.60 | 0.45 |
| All Datasets + BoolQ | 0.73 | 0.84 | 0.82 | 0.83 |

*Table 5.* Accuracy of LLMs with Different Contexts

