# OpenReview forum: "Fast Training Dataset Attribution via In-Context Learning"
_ICML.cc/2024/Workshop/ICL — ICML 2024 Workshop ICL Poster_

### Official Review · Reviewer_v2Cw · 2024-06-08
**Novel solutions but the evaluation lacks conviction.**

**Rating:** 2
**Fit:** 3
**Confidence:** 2

**Workshop Review:**

This work studies how to measure the contributions of training data to the outputs of aligned LLMs, and thereafter proposes two estimate methods from novel perspectives, one is similarity-based and the other one is a mixture distribution model approach. The authors valid the effectiveness of proposed methods among four LLMs across different sizes.

Pros:

- To the best of my knowledge, the solutions are novel for this problem setting.
- The results and analysis provide multiple useful indications to the properties of LLMs, including the exposure to data from target domain, the impact of in-context demonstrations on outputs.

Cons:

- The evaluation is a bit flawed. It only tests on the BoolQ Q&A dataset.

**Reason For Not Giving Higher Score:**

The evaluation is insufficient.

**Reason For Not Giving Lower Score:**

The proposed solutions are novel, and using ICL to estimate the contributions of training data to output can avoid demanding fine-tuning.

---

### Official Review · Reviewer_paNz · 2024-06-09

**Rating:** 2
**Fit:** 3
**Confidence:** 2

**Workshop Review:**

This paper investigates an interesting approach to data attribution using in-context learning. The authors relates the data attribution problem to the change in LLM's output when the relevant training data is present in a model's context. The proposed technique appears novel and involves creative use of in-context learning, thus relevant to the ICL workshop.

However, the paper leaves quite a few important assumptions untested, for example one of the foundational assumption of this work is
> providing a dataset as context to an LLM
> trained on that dataset changes its output less compared
> to when the LLM was not trained on the dataset

Furthermore, the evaluation protocol is not clean and potentially weak. The author uses subject-level partitions (Law, Biology, History, etc) of wikipedia dataset as candidates for data attribution. However, when evaluating on BoolQ dataset, the author included the context of BoolQ dataset as one of the attribution candidate too, and unsurprisingly most of LLM's output are attributable to BoolQ's context. A stronger evaluation would be to use something like TriviaQA's Wikipedia partition, and try attributing answers to each partition of Wikipedia dataset.

The clarity of the paper can be improved, examples of clarity issues include:
> L55:  Given the modularity of LLM structures, this assumption is not fully realistic
  I do not understand how LLM's structure can be described as "modular", neither do I understand why this has anything to do with the assumption that you can model LM probability as a sum of component LM probabilities, each corresponding to a dataset.

**Reason For Not Giving Higher Score:**

weak evaluation, sometimes confusing writing

**Reason For Not Giving Lower Score:**

the authors propose a novel technique to address an important problem

---

### Meta-Review · Area_Chair_Ppap · 2024-06-14

**Recommendation:** 2

**Metareview:**

The paper explores using in-context learning and prompt engineering to estimate how training data influences the outputs of large language models (LLMs). It introduces two methods and finds that the mixture distribution model approach is more robust to retrieval noise and provides more reliable data contribution estimates.

Both reviewers agree on the fit and the overall score of the paper. However, both reviewers point out flaws in the experimental setup of the current paper which should be addressed by the authors.

I recommend acceptance as poster.

---

### Decision · Program_Chairs · 2024-06-17

**Decision:**

Accept (Poster)

**Comment:**

**Accept with minor revision**: Please address the concerns of the reviewers about your experimental setup.